# SALR: Sharpness-aware Learning Rates for Improved Generalization

## Abstract

In an effort to improve generalization in deep learning, we propose SALR: a sharpness-aware learning rate update technique designed to recover flat minimizers. Our method dynamically updates the learning rate of gradient-based optimizers based on the local sharpness of the loss function. This allows optimizers to automatically increase learning rates at sharp valleys to increase the chance of escaping them. We demonstrate the effectiveness of SALR when adopted by various algorithms over a broad range of networks. Our experiments indicate that SALR improves generalization, converges faster, and drives solutions to significantly flatter regions.

## 1 Introduction

Generalization in deep learning has recently been an active area of research. The efforts to improve generalization over the past two decades have brought upon many cornerstone advances and techniques; be it dropout (Gal & Ghahramani, 2016), batch-normalization (Ioffe & Szegedy, 2015), data-augmentation (Shorten & Khoshgoftaar, 2019), weight decay (Loshchilov & Hutter, 2019), adaptive gradient-based optimization (Kingma & Ba, 2015), architecture design and search (Radosavovic et al., 2020), ensembles and their Bayesian counterparts (Garipov et al., 2018; Izmailov et al., 2018), amongst many others. Yet, recently, researchers have discovered that the concept of sharpness/flatness plays a fundamental role in generalization. Though sharpness was first discussed in the context of neural networks in the early work of Hochreiter & Schmidhuber (1997), it was only brought to the forefront of deep learning research after the seminal paper by Keskar et al. (2017). While trying to investigate decreased generalization performance when large batch sizes are used (LeCun et al., 2012) in stochastic gradient descent (SGD), Keskar et al. (2017) notice that this phenomena can be justified by the ability of smaller batches to reach flat minimizers. Such flat minimizers in turn, generalize well as they are robust to low precision arithmetic or noise in the parameter space (Dinh et al., 2017; Kleinberg et al., 2018), as shown in Figure 1.

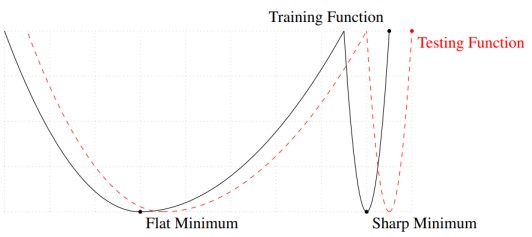

Figure 1: A conceptual sketch of flat and sharp minima (Keskar et al., 2017).

Since then, the generalization ability of flat minimizers has been repeatedly observed in many recent works (Neyshabur et al., 2017a; Goyal et al., 2017; Li et al., 2018; Izmailov et al., 2018). Indeed, flat minimizers can potentially tie together many of the aforementioned approaches aimed at generalization. For instance, (1) higher gradient variance, when batches are small, increases the probability to avoid sharp regions (same can be said for SGD compared to GD) (Kleinberg et al., 2018) (2) averaging over multiple hypotheses leads to wider optima in ensembles and Bayesian deep learning (Izmailov et al., 2018) (3) regularization techniques such as dropout or over-parameterization can adjust the loss landscape into one that allows first order methods to favor wide valleys (Chaudhari et al., 2019; Allen-Zhu et al., 2019).

In this paper we study the direct problem of developing an algorithm that can converge to flat minimizers. Specifically, we introduce SALR: a sharpness aware learning rate designed to explore the

loss-surface of an objective function and avoid undesired sharp local minima. SALR dynamically updates the learning rate based on the sharpness of the neighborhood of the current solution. The idea is simple: automatically increase the learning rates at relatively sharp valleys in an effort to escape them. One of the key features of SALR is that it can be fitted into any gradient based method such as Adagrad (Duchi et al., 2011), ADAM (Kingma & Ba, 2015) and also into recent approaches towards escaping sharp valleys such as Entropy-SGD (Chaudhari et al., 2019).

## 1.1 RELATED WORK

From a theoretical perspective, generalization of deep learning solutions has been explained through multiple lenses. One of which is uniform stability (Bottou & Le Cun, 2005; Bottou & Bousquet, 2008; Hardt et al., 2016; Gonen & Shalev-Shwartz, 2017; Bottou et al., 2018). An algorithm is uniformly stable if for all data sets differing in only one element, nearly the same outputs will be produced (Bousquet & Elisseeff, 2002). Hardt et al. (2016) show that SGD satisfies this property and derive a generalization bound for models learned with SGD. From a different viewpoint, Choromanska et al. (2015); Kawaguchi (2016); Poggio et al. (2017); Mohri et al. (2018) attribute generalization to the complexity of the hypothesis-space. Using measures like Rademacher complexity (Mohri & Rostamizadeh, 2009) and the Vapnik-Chervonenkis (VC) dimension (Sontag, 1998), the former works show that deep hypothesis spaces are typically more advantageous in representing complex functions. Besides that, the importance of flatness on generalization has been theoretically highlighted through PAC-Bayes bounds (Dziugaite & Roy, 2017; Neyshabur et al., 2017b; Wang et al., 2018). These papers highlight the ability to derive non-vacuous generalization bounds based on the sharpness of a model class while arguing that relatively flat solutions yield tight bounds.

From an algorithmic perspective, approaches to recover flat minima are still limited. Most notably, Chaudhari et al. (2019) developed the Entropy-SGD algorithm. Entropy-SGD defines a local-entropy-based objective which smoothens the energy landscape based on its local geometry. This in turn allows SGD to attain flatter solutions. Indeed, this approach was motivated by earlier work in statistical physics (Baldassi et al., 2015; 2016) which proves the existence of non-isolated solutions that generalize well in networks with discrete weights. Such non-isolated solutions correspond to flat minima in continuous settings. The authors then propose a set of approaches based on ensembles and replicas of the loss to favor wide solutions. Not too far, recent methods in Bayesian deep learning (BDL) have also shown potential to recover flat minima. BDL basically averages over multiple hypotheses weighted by their posterior probabilities (ensembles being a special case of BDL (Izmailov et al., 2018)). One example, is the stochastic weighted averaging (SWA) algorithm proposed by Izmailov et al. (2018). SWA simply averages over multiple points along the trajectory of SGD to potentially find flatter solutions compared to SGD. Another example is the SWA-Gaussian (SWAG). SWAG defines a Gaussian posterior approximation over neural network weights. Afterwards, samples are taken from the approximated distribution to perform Bayesian model averaging (Maddox et al., 2019).

Here we also note the recent work by Patel (2017) which partially motivates our method. Upon the aforementioned observations in Keskar et al. (2017), Patel (2017) shows that the learning rate lower-bound threshold for the divergence of batch SGD, run on quadratic optimization problems, increases for larger batch-sizes. In general non-convex settings, given a problem with $N$ local minimizers, one can compute $N$ lower bound thresholds for local divergence of batch SGD. The number of minimizers for which batch SGD can converge is non-decreasing in the batch size. This is used to explain the tendency of low-batch SGD to converge to flatter minimizers compared to large-batch SGD. The former result links the choice of batch size and its effect on generalization to the choice of the learning rate. With the latter being a tunable parameter, to our knowledge, developing a dynamic choice of the learning rate that targets convergence to flat minimizers has not been studied before.

## 2 GENERAL FRAMEWORK

In this paper, we propose a framework that dynamically chooses a *Sharpness-Aware Learning Rate* to promote convergence to flat minimizers. More specifically, our proposed method locally approximates sharpness at the current iterate and dynamically adjusts the learning rate accordingly. In sharp regions, relatively large learning rates are attained to increase the chance of escaping that region. In contrast, when the current iterate belongs to a flat region, our method returns a relatively small

learning rate to guarantee convergence. Our framework can be adopted by any local search descent method and is detailed in Algorithm 1.

---

**Algorithm 1:** Sharpness-Aware Learning Rate (SALR) Framework

---

**Input:** Starting point $\boldsymbol{\theta}_0$, initial learning rate $\eta_0$, Number of iterations $K$.

**for** $k = 0, 1, \ldots, K$ **do**

    Estimate $\widehat{S}_k$, the local sharpness around the current iterate $\boldsymbol{\theta}_k$;

    Set $\eta_k = \eta_0 \dfrac{\widehat{S}_k}{\text{Median}\left\{\widehat{S}_i\right\}_{i=1}^{k}}$;

    Compute $\boldsymbol{\theta}_{k+1}$ using some local search descent method (Gradient Descent, Stochastic Gradient Descent, ADAM, ...);

**end**

Return $\boldsymbol{\theta}_K$.

---

As detailed in Algorithm 1, at every iterate $k$, we compute the learning rate as a function of the local sharpness parameters $\left\{\widehat{S}_k\right\}_{i=1}^{k}$. The main intuition is to have the current learning rate to be an increasing function of the current estimated sharpness. Since the scale of the sharpness at different points can vary when using different networks or datasets (Dinh et al., 2017), we normalize our estimated sharpness by dividing by the median of the sharpness of previous iterates. For instance in Figure 2, despite having a similar sharpness measure, we consider the minimizer around $\boldsymbol{\theta} = 1$ to be sharp relative to the blue plot and flat relative to red plot. Normalization resolves this issue by helping our sharpness measure attain scale invariant properties.

One can think of the median of previous sharpness values as a global sharpness parameter the algorithm is trying to learn. When $k$ is sufficiently large, the variation in the global sharpness parameter among different iterates will be minimal. From an algorithmic perspective, SALR exploits a neighborhood around the current iterate to dynamically compute a desired learning rate while simultaneously exploring the sharpness of the landscape to refine this global sharpness parameter.

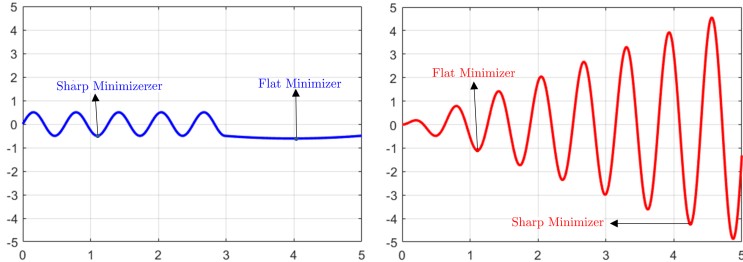

Figure 2: Sharp/Flat minimizers relative to the landscape.

## 3 SHARPNESS MEASURE

Several sharpness/flatness measures have been defined in recent literature (Rangamani et al., 2019; Keskar et al., 2017; Hochreiter & Schmidhuber, 1997). For instance, Hochreiter & Schmidhuber (1997) computes flatness by measuring the size of the connected region in the parameter space where the objective remains approximately constant. In a more recent paper, Rangamani et al. (2019) proposed a scale invariant flatness measure based on the quotient manifold. Computing such notions for complex non-convex landscapes can be intractable in practice. In addition to the cited results, Keskar et al. (2017) quantifies flatness by finding the difference between the maximum value of the loss function within a small neighborhood around a given point and the current value. More specifically, they define sharpness as follows:

$$\phi(\varepsilon, \boldsymbol{\theta}) \triangleq \frac{S(\varepsilon, \boldsymbol{\theta})}{1 + f(\boldsymbol{\theta})} \qquad \text{and} \qquad S(\varepsilon, \boldsymbol{\theta}) = \max_{\boldsymbol{\theta}' \in \mathbb{B}_\varepsilon(\boldsymbol{\theta})} f(\boldsymbol{\theta}') - f(\boldsymbol{\theta}), \qquad (1)$$

where $\mathbb{B}_\varepsilon(\boldsymbol{\theta})$ is a euclidean ball with radius $\varepsilon$ centered at $\boldsymbol{\theta}$ and $1 + f(\boldsymbol{\theta})$ is a normalizing coefficient . One drawback of equation 1 is that the sharpness value around a maximizer is nearly zero. To resolve

this issue, one can simply modify the sharpness measure in equation 1 as follows:

$$S(\varepsilon, \boldsymbol{\theta}) \triangleq \max_{\boldsymbol{\theta}' \in \mathbb{B}_\varepsilon(\boldsymbol{\theta})} f(\boldsymbol{\theta}') - \min_{\boldsymbol{\theta}' \in \mathbb{B}_\varepsilon(\boldsymbol{\theta})} f(\boldsymbol{\theta}'). \tag{2}$$

It can be easily shown that if $\boldsymbol{\theta}$ is a local minimizer, equation 2 is equivalent to equation 1. Both measures defined in equation 1 and equation 2 require solving a possibly non-convex function which is in general NP-Hard. For computational feasibility, we provide a sharpness approximation by running $n_1$ gradient ascent and $n_2$ gradient descent steps. The resulting solutions are used to approximate the maximization and minimization optimization problems. Here we note that our definition for sharpness does not include a normalizing coefficient, as median $\left\{\widehat{S}_i\right\}_{i=1}^k$ in Algorithm 1 plays this role. The details of the approximation are shown in the Definition 1.

**Definition 1.** *Given $\boldsymbol{\theta} \in \mathbb{R}^n$, iteration numbers $n_1$ and $n_2$, and step-size $\gamma$, we define the sharpness measure*

$$\widehat{S}(\boldsymbol{\theta}) \triangleq f(\boldsymbol{\theta}_{k,+}^{(n_2)}) - f(\boldsymbol{\theta}_k) + f(\boldsymbol{\theta}_k) - f(\boldsymbol{\theta}_{k,-}^{(n_1)}) + = f(\boldsymbol{\theta}_{k,+}^{(n_2)}) - f(\boldsymbol{\theta}_{k,-}^{(n_1)}),$$

*where $\boldsymbol{\theta}_{k,+}^{(0)} = \boldsymbol{\theta}_{k,-}^{(0)} = \boldsymbol{\theta}_k$,*

$$\boldsymbol{\theta}_{k,-}^{(n_1)} = \boldsymbol{\theta}_k - \sum_{i=0}^{n_1-1} \gamma \frac{\nabla f(\boldsymbol{\theta}_{k,-}^{(i)})}{\left\|\nabla f(\boldsymbol{\theta}_{k,-}^{(i)})\right\|}, \qquad and \qquad \boldsymbol{\theta}_{k,+}^{(n_2)} = \boldsymbol{\theta}_k + \sum_{i=0}^{n_2-1} \gamma \frac{\nabla f(\boldsymbol{\theta}_{k,+}^{(i)})}{\left\|\nabla f(\boldsymbol{\theta}_{k,+}^{(i)})\right\|}.$$

**Remark 2.** *In contrast to the measures defined in equation 2 and equation 1, Definition 1 does not require a ball radius $\varepsilon$. However, our definition requires specifying the step-size $\gamma$ and the number of ascent and descent iterations.*

**Remark 3.** *Running gradient descent/ascent with fixed step-size near a minimizer can return a very small sharpness value even if the minimizer is sharp. This is due to the small gradient norm around a minimizer. To resolve this issue, we normalize the gradient at every descent/ascent step. Moreover, normalizing by the norm of the gradient helps in understanding the radius of the ball containing the iterates $\{\boldsymbol{\theta}_{k,-}^{(j)}\}_{j=1}^{n_1}$ and $\{\boldsymbol{\theta}_{k,+}^{(j)}\}_{j=1}^{n_2}$.*

Figure 3 shows the plots of the three different sharpness measures defined in this section when computed for a function $f(\theta) = 0.5\theta\sin(3\theta) + 1$. Notice that the blue plot corresponding to the sharpness measure $\phi(\cdot)$ attains a zero value at local maximizers compared to a positive value for the other two sharpness plots. Moreover, notice that the sharpness value in these three plots attains a small value near the local minimizer. This can be explained by our choice of radius $\varepsilon = 0.1$ which limits the neighborhood being exploited. Increasing the radius for $\phi(\varepsilon, \cdot)$ and $S(\varepsilon, \cdot)$ (increasing $n_1$ and $n_2$ for $\widehat{S}$) will provide higher values around the minimizer. We next show that using our sharpness measure in Definition 1, gradient descent with SALR framework in Algorithm 1, denoted as GD-SALR, escapes sharp local minima.

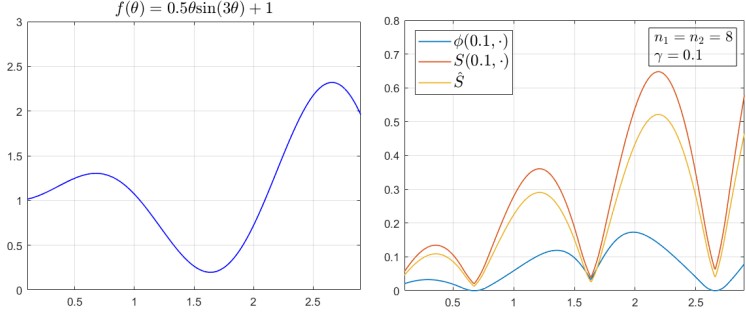

Figure 3: Sharpness measure plots for $\phi$, $S$, and $\widehat{S}$ on function $f$.

## 4 THEORETICAL RESULTS

In this section, we focus on analyzing the convergence of vanilla GD when adopting our sharpness-aware learning rate framework in Algorithm 1. We show that GD-SALR escapes any given neighborhood of a sharp local minimum by choosing a sufficiently large step-size. Throughout this section,

we make the following assumptions that are standard for the convergence theory of gradient descent methods.

**Assumption 1.** *The objective function $f$ is twice continuously differentiable and $L_0$-Lipschitz continuous. The gradient function $\nabla f(\cdot)$ is L-Lipschitz continuous with Lipschitz constant $L$. Furthermore, the gradient norm is bounded, i.e. there exists a scalar constant $g_{max} > 0$ such that $\|\nabla f(\boldsymbol{\theta})\| \leq g_{max}$ for all $\boldsymbol{\theta}$.*

The next theorem shows that GD with a sufficiently large step-size escapes a given strongly convex neighborhood around a local minimum $\boldsymbol{\theta}^*$.

**Theorem 4.** *Suppose that $f$ is $\mu$-strongly convex function in a neighborhood $\mathbb{B}_\delta(\boldsymbol{\theta}^*)$ around a local minimum $\boldsymbol{\theta}^*$, i.e. $\lambda_{min}\left(\nabla^2 f(\boldsymbol{\theta})\right) \geq \mu$ for all $\boldsymbol{\theta} \in \mathbb{B}_\delta(\boldsymbol{\theta}^*) \triangleq \{\boldsymbol{\theta} \mid \|\boldsymbol{\theta} - \boldsymbol{\theta}^*\|_2 \leq \delta\}$. Running vanilla GD with $\boldsymbol{\theta}_0 \in \mathbb{B}_\delta(\boldsymbol{\theta}^*)$ and learning rate $\eta_k \geq \dfrac{2 + \varepsilon}{\mu}$ for some fixed $\varepsilon > 0$, there exists $\widehat{k}$ with $\boldsymbol{\theta}_{\widehat{k}} \notin \mathbb{B}_\delta(\boldsymbol{\theta}^*)$.*

*Proof.* The proof of lemma is relegated to the Appendix B. $\qquad\square$

Our next result shows that GD-SALR escapes sharp local minima by dynamically choosing a sufficiently large step-size in a local strongly convex region.

**Theorem 5.** *Suppose that $f$ is a $\mu$-strongly convex function in a neighborhood $\mathbb{B}_\delta(\boldsymbol{\theta}^*)$ around a local minimum $\boldsymbol{\theta}^*$, i.e. $\lambda_{min}\left(\nabla^2 f(\boldsymbol{\theta})\right) \geq \mu$ for all $\boldsymbol{\theta} \in \mathbb{B}_\delta(\boldsymbol{\theta}^*) \triangleq \{\boldsymbol{\theta} \mid \|\boldsymbol{\theta} - \boldsymbol{\theta}^*\|_2 \leq \delta\}$. Under Assumption 1, run GD-SALR (Gradient descent with step size choice according to Algorithm 1 and Definition 1) with*

$$n_1 \geq \frac{a_1}{\left(\log\left(1 + \dfrac{\mu\, g_{min}}{L\, g_{max} - \mu\, g_{min}}\right)\right)^2} - \frac{1}{a_1}, \quad n_2 \frac{a_2}{\left(\log\left(1 + \dfrac{\mu\, g_{min}}{L\, g_{max}}\right)\right)^2} - \frac{1}{a_2}, \quad and \quad \gamma = \frac{g_{min}}{L},$$

*where*

$$a_1 = \frac{(2 + \epsilon)L_0}{\eta_0(g_{max}L - g_{min}\mu)} + \frac{g_{min}\mu}{2(g_{max}L - g_{min}\mu)}, \quad and \quad a_2 = \frac{(2 + \epsilon)L_0}{\eta_0 g_{max}L} + \frac{\mu^2\, g_{min}}{2L^2 g_{max}}, \quad \epsilon, \eta_0 > 0,$$

*and $g_{min} > 0$ is a lower bound that satisfies*

$$\max\left\{\left\|\nabla f\left(\boldsymbol{\theta}_{k,-}^{(n_1-1)}\right)\right\|, \|\nabla f(\boldsymbol{\theta}_k)\|\right\} \geq g_{min}.$$

*If $\delta > \max\{n_1, n_2\}\gamma$, then there exists $\widehat{k}$ with $\boldsymbol{\theta}_{\widehat{k}} \notin \mathbb{B}_\delta(\boldsymbol{\theta}^*)$.*

*Proof.* The proof of the theorem is relegated to the Appendix C. $\qquad\square$

**Remark 6.** *In the context of machine learning, our theorem shows that our algorithm can potentially escape sharp regions even when all the data are used (full-batch). In their work, Patel (2017) show that when using large batch sizes we require a higher learning rate to escape sharp minima. This provides an insight on the favorable empirical results presented in Section 6 when running SGD-SALR. Moreover, our dynamic choice of high learning rate in sharp regions can potentially allow running SGD with larger batch sizes while still escaping sharp minimizers. This in turn provides an avenue for improved parallelism (Dean et al., 2012; Das et al., 2016).*

In the next section, we generalize our proposed framework to the stochastic setting.

## 5 STOCHASTIC APPROXIMATION OF SHARPNESS

The concept of generalization is more relevant when solving problems arising in machine learning settings. Under the empirical risk minimization framework, the problem of training machine learning models can be mathematically formulated as the following optimization problem

$$\min_{\boldsymbol{\theta} \in \mathbb{R}^n} f(\boldsymbol{\theta}) \triangleq \frac{1}{m} \sum_{i=1}^m f_i(\boldsymbol{\theta}), \tag{3}$$

where $f_i$ is a loss function parameterized with parameter $\boldsymbol{\theta}$ corresponding to data point $i \in \{1, 2, \ldots, m\}$. The most popular algorithm used to solve such optimization problems is the stochastic gradient descent which iteratively updates the parameters using the following update rule:

$$\boldsymbol{\theta}_{k+1} = \boldsymbol{\theta}_k - \eta_k \left( \frac{1}{|B_k|} \sum_{i \in B_k} \nabla f_i(\boldsymbol{\theta}_k) \right),$$

where $B_k$ is the batch sampled at iteration $k$ and $\eta_k$ is the learning rate. To apply our framework in stochastic settings, we provide a stochastic procedure for computing the sharpness measure at a given iterate. Details are provided in Algorithm 2. By adopting Algorithm 2, our framework can be applied to numerous popular algorithms like SGD, ADAM and Entropy-SGD. The detailed implementation of SGD-SALR and ADAM-SALR can be found in Algorithms 3 and 4 in Appendix A.

---

**Algorithm 2:** Calculating stochastic sharpness at iteration $k$

**Data:** batch size $B_k$, base learning rate $\gamma$, current iterate $\boldsymbol{\theta}_k$, iteration number $n_1, n_2$

Set $\boldsymbol{\theta}_{k,+}^{(0)} = \boldsymbol{\theta}_{k,-}^{(0)} = \boldsymbol{\theta}_k$;

**for** $i = 0 : n_1 - 1$ **do**

$$\boldsymbol{\theta}_{k,-}^{(i+1)} = \boldsymbol{\theta}_{k,-}^{(i)} - \gamma \left( \frac{1}{|B_k|} \sum_{j \in B_k} \frac{\nabla f_j\left(\boldsymbol{\theta}_{k,-}^{(i)}\right)}{\left\|\nabla f_j\left(\boldsymbol{\theta}_{k,-}^{(i)}\right)\right\|} \right);$$

**end**

**for** $i = 0 : n_2 - 1$ **do**

$$\boldsymbol{\theta}_{k,+}^{(i+1)} = \boldsymbol{\theta}_{k,+}^{(i)} + \gamma \left( \frac{1}{|B_k|} \sum_{j \in B_k} \frac{\nabla f_j\left(\boldsymbol{\theta}_{k,+}^{(i)}\right)}{\left\|\nabla f_j\left(\boldsymbol{\theta}_{k,+}^{(i)}\right)\right\|} \right);$$

**end**

Return $\hat{S}_k = \dfrac{1}{|B_k|} \sum_{j \in B_k} f_j\left(\boldsymbol{\theta}_{k,+}^{(n_2)}\right) - \dfrac{1}{|B_k|} \sum_{j \in B_k} f_j\left(\boldsymbol{\theta}_{k,-}^{(n_1)}\right)$;

---

## 6 EMPIRICAL RESULTS

In this section, we present experimental results on image classification and text prediction datasets. We show that our framework SALR can be adopted by many optimization methods and achieve notable improvements over a broad range of networks. We compare SALR with Entropy-SGD (Chaudhari et al., 2019) and SWA (Izmailov et al., 2018). Besides those benchmarks, we also use the conventional SGD and ADAM (Kingma & Ba, 2015) as baseline references. All aforementioned methods are trained with batch normalization (Ioffe & Szegedy, 2015) and dropout of probability 0.5 after each layer (Gal & Ghahramani, 2016). We replicate each experiment 30 times to obtain the mean and standard deviation of testing errors. In our experiments, we do not tune any hyperparameters. We consider some typical networks such as mnistfc (Ioffe & Szegedy, 2015), ResNet (He et al., 2016), DenseNet (Iandola et al., 2014), MobileNetV2 (Sandler et al., 2018) and RegNetX (Radosavovic et al., 2020).

### 6.1 MNIST/CIFAR-10

We run Algorithm 3 SGD-SALR for 20 epochs. We collect the sharpness measure every $c = 2$ iterations and set $n_1 = n_2 = 5$ as detailed in Algorithm 2. The experimental settings for other benchmark models are as follows: (1) **SGD:** we run SGD for 100 epochs using decay learning rates. (2) **SWA:** the setting is the same as SGD. In the SWA stage, we switch to a cyclic learning rate schedule as suggested in Izmailov et al. (2018). (3) **Entropy-SGD:** following the setting in Chaudhari et al. (2019), we train Entropy-SGD for 20 epochs and set Langevin iterations $L_a = 5$. (4) **Entropy-SGD-SALR:** the setting is same as Entropy-SGD, however, we update the learning rate of Entropy-SGD using Algorithm 2. Further details on each benchmark setting can be found in Appendix D. The results are reported in Table 1.

| Network | SGD | SWA | Entropy-SGD | Entropy-SGD-SALR | SGD-SALR |
|---|---|---|---|---|---|
| ResNet18 | 98.97 (0.01) | 98.99 (0.01) | 99.11 (0.01) | **99.19** (0.01) | 99.21 (0.03) |
| RegNetX | 99.02 (0.03) | 98.95 (0.01) | 99.20 (0.03) | **99.47** (0.01) | 99.45 (0.01) |
| LeNet | 98.50 (0.01) | 98.69 (0.01) | 99.05 (0.03) | 99.16 (0.01) | **99.17** (0.01) |
| MobileNetV2 | 98.79 (0.02) | 98.80 (0.01) | 99.26 (0.00) | 99.44 (0.01) | **99.50** (0.01) |
| mnistfc | 98.02 (0.02) | 98.21 (0.02) | 98.58 (0.01) | 99.00 (0.01) | **99.02** (0.01) |

Table 1: Classification accuracy on MNIST

We increase the number of training epochs for Entropy-SGD/Entropy-SGD-SALR/SGD-SALR to 40 and the training epochs of SGD/SWA to 200. Experimental results are reported in Table 2.

| Network | SGD | SWA | Entropy-SGD | Entropy-SGD-SALR | SGD-SALR |
|---|---|---|---|---|---|
| ResNet18 | 88.17 (0.04) | 88.56 (0.01) | 91.05 (0.01) | 91.21 (0.01) | **91.21** (0.02) |
| ResNet50 | 88.44 (0.03) | 88.83 (0.01) | 90.80 (0.03) | 92.27 (0.01) | **92.44** (0.01) |
| All-CNN-BN | 91.93 (0.01) | 92.20 (0.01) | 91.13 (0.01) | 92.16 (0.01) | **92.45** (0.05) |
| ResNet101 | 95.11 (0.02) | 95.56 (0.01) | 95.51 (0.01) | 95.87 (0.01) | **95.99** (0.00) |
| RegNetX | 94.24 (0.02) | 94.23 (0.01) | 94.26 (0.01) | 95.00 (0.01) | **95.01** (0.01) |

Table 2: Classification accuracy on CIFAR10

To illustrate the flexibility of our framework, we change the base optimizer SGD to ADAM and re-run all the experiments under a similar setting as that of Table 2. Results are reported in Table 3. Finally, in Table 4, we report the sharpness measure of the final solution obtained by each optimization approach. We also conduct some experiments on CIFAR-100. Results are deferred to Appendix.

| Network | ADAM | SWA | Entropy-ADAM | Entropy-ADAM-SALR | ADAM-SALR |
|---|---|---|---|---|---|
| ResNet18 | 88.01 (0.01) | 88.43 (0.01) | 91.06 (0.01) | 91.17 (0.01) | **91.23** (0.01) |
| ResNet50 | 87.98 (0.02) | 88.41 (0.03) | 90.61 (0.01) | 92.29 (0.01) | **92.31** (0.01) |
| All-CNN-BN | 91.95 (0.01) | 92.27 (0.01) | 91.10 (0.01) | 92.15 (0.01) | **92.35** (0.01) |
| ResNet101 | 95.00 (0.01) | 95.57 (0.01) | 95.56 (0.01) | **96.03** (0.01) | 95.97 (0.00) |
| RegNetX | 94.33 (0.01) | 94.12 (0.01) | 94.21 (0.01) | **95.06** (0.01) | 95.02 (0.01) |

Table 3: Classification accuracy on CIFAR10 using ADAM

## 6.2 TEXT PREDICTION

We train an LSTM network on the Penn Tree Bank (PTB) dataset for word-level text prediction. This dataset contains about one million words. Following the guideline in [1] and [2], we train PTB-LSTM with 66 million weights. SGD and SWA are trained with 55 epochs. Entropy-SGD ($L = 5$) and SALR ($c = 2$) are trained with 11 epochs. Overall, **all methods have the same number of gradient calls (i.e., wall-clock times).** We then train an LSTM to perform character-level text-prediction using War and Peace (WP). We follow the procedures in [2] and [3]. We train Adam/SWA and Entropy-SGD/SALR with 50 and 10 epochs, respectively. We report the perplexity on the test set in Table 5.

## 6.3 ANALYSIS

Based on Tables 1-5, we can obtain some important insights. First, methods adopting SALR show superior performance over their benchmarks. This increase in classification accuracy (or decrease in perplexity) is consistent across both datasets and a range of network structures. We also observed that SGD-SALR tends to outperform other SALR based methods in most settings while achieving comparable results in others. Second, and more interestingly, this superior performance is achieved with 5 times less epochs compared to SGD, ADAM and SWA. The caveat however is that SALR and Entropy respectively require $(n_1 + n_2)/c = 5$ and $L_a = 5$ more gradient calls at each iteration, hence making the total computational needs the same as ADAM, SGD and SWA. Third and as shown in Table 4, it is clear that SALR drives solutions to significantly flatter regions. This highlights the effectiveness of dynamically adjusting learning rates based on the relative sharpness of the current

| $\times 10^{-3}$ | SGD | SWA | Entropy SGD | Entropy-SGD SALR | SGD SALR |
|---|---|---|---|---|---|
| ResNet18 | 3.23 (0.47) | 3.14 (0.51) | 1.06 (0.41) | 1.04 (0.40) | **0.99** (0.36) |
| ResNet50 | 8.11 (0.52) | 7.65 (0.33) | 3.55 (0.44) | 3.63 (0.31) | **3.22** (0.63) |
| All-CNN-BN | 11.02 (1.00) | 10.65 (1.21) | **6.12** (0.88) | 6.35 (0.84) | 6.30 (0.91) |
| ResNet101 | 7.00 (0.87) | 6.91 (0.22) | 5.33 (0.60) | **6.07** (0.82) | 6.13 (0.70) |
| RegNetX | 9.56 (1.03) | 9.66 (0.69) | 9.41 (0.50) | **8.77** (0.68) | 8.90 (0.74) |

Table 4: Sharpness of final solutions (CIFAR-10, SGD)

| PTB | SGD | SWA | Entropy-SGD | SALR |
|---|---|---|---|---|
| PTB-LSTM | 78.4 (0.22) | 78.1 (0.25) | 72.15 (0.16) | 71.42 (0.14) |
| WP-LSTM | 1.223 (0.01) | 1.220 (0.05) | 1.095 (0.01) | 1.089 (0.02) |

Table 5: Perplexity on PTB/WP

iterate. To further demonstrate the advantage of SALR framework, we plot the testing error curves for SGD, Entropy-SGD, Entropy-SALR and SALR in Figure 4 (Left). Interestingly, the overall trend when adding SALR to SGD is drastically changed with the trend smoothly and more consistently increasing its performance. This can be potentially explained through the capability of SALR to quickly escape sharp regions relative to SGD and hence attain larger and more consistent rates of improvement across Epochs. We also plot the change of sharpness/learning rates over some SALR iterations in the last Epoch in Figure 4 (Right). This figure highlights the dynamics of both learning rates and sharpness, highlighting that the median sharpness tends to stabilize and hence leading to a proportional relationship between learning rates and sharpness.

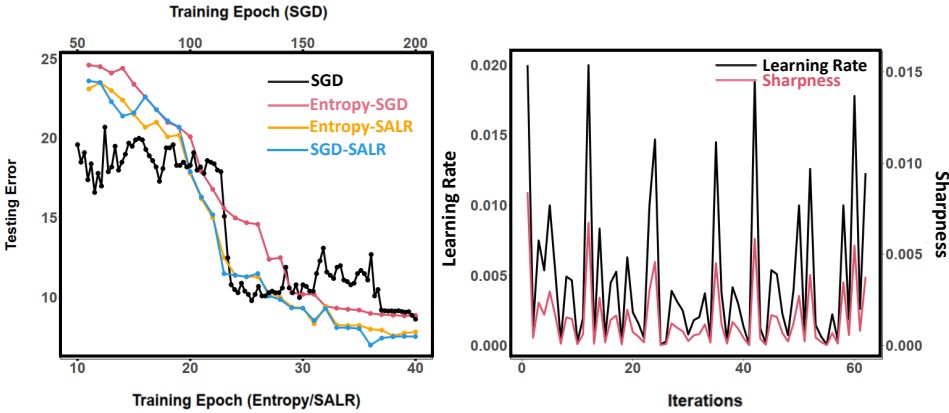

Figure 4: (Left) All-CNN-BN: Change of Testing Errors over Epochs. (Right) Change of sharpness and learning rate over iterations.

## 7 DISCUSSION & OPEN PROBLEMS

In this paper we introduce SALR: an optimization tool that aims to recover flat minima through dynamically updating the learning rate based on the current solution's relative sharpness. SALR can be readily plugged into any gradient based method. Experiments show that SALR can deliver promising improvements over a range of optimization methods and network structures. In light of this work, we hope researchers further investigate landscape dependant learning rates as they can offer a potential alternative/unifying framework for many aforementioned attempts to achieve improved generalization. Also, quasi-Newton approximations of $S_k$ instead of first order methods and generalization bounds for SALR remain open problems worth investigating.

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

# Supplementary Material

## A  EXAMPLES OF SALR-BASED ALGORITHMS

The SALR framework can be fitted into many optimization algorithms. In this section we provide some examples. In Algorithm 3 and 4, we present the SGD-SALR algorithm and the ADAM-SALR algorithm, respectively.

---

**Algorithm 3:** The SGD-SALR Algorithm

---

**Data:** base learning rate $\eta_0$, number of iterations $K$, frequency $c$, initial weight $\boldsymbol{\theta}_0$
**Result:** weight vector $\boldsymbol{\theta}$.
Set $\mathcal{S} = \emptyset$;
**for** $k = 0 : K$ **do**
    **if** $k \mod c = 0$ **then**
        Calculate $\widehat{S}_k$ using Algorithm 2;
    **end**
    Compute $\mathcal{S} = \text{Median}(\{\widehat{S}_k\})$
    Set $\eta_k = \eta_0 \dfrac{\widehat{S}_k}{\mathcal{S}}$;
    $\boldsymbol{\theta}_{k+1} = \boldsymbol{\theta}_k - \eta_k \dfrac{1}{|B_k|} \sum_{j \in B_k} \nabla f_j(\boldsymbol{\theta}_k)$;
**end**
Set $\boldsymbol{\theta} = \boldsymbol{\theta}_K$;
Return $\boldsymbol{\theta}$.

---

**Algorithm 4:** The ADAM-SALR Algorithm

---

**Data:** base learning rate $\eta_0$, exponential decay rates for the moment estimates $\beta_1, \beta_2 \in [0, 1)$,
        number of iterations $K$, frequency $c$, initial weight $\boldsymbol{\theta}_0$, perturbation $\epsilon$
**Result:** weight vector $\boldsymbol{\theta}$.
Set $\mathcal{S} = \emptyset$;
Set $m_0 = v_0 = 0$;
**for** $k = 0 : K$ **do**
    **if** $k \mod c = 0$ **then**
        Calculate $\widehat{S}_k$ using Algorithm 2;
    **end**
    Compute $\mathcal{S} = \text{Median}(\{\widehat{S}_k\})$
    Set $\eta_k = \eta_0 \dfrac{\hat{S}_k}{\mathcal{S}}$;
    $g = \dfrac{1}{|B_k|} \sum_{j \in B_k} \nabla f_j(\boldsymbol{\theta}_k)$
    $m_{k+1} = \beta_1 m_k + (1 - \beta_1) g$
    $v_{k+1} = \beta_2 v_k + (1 - \beta_2) g^2$
    $\widehat{m}_{k+1} = m_k / (1 - \beta_1^{k+1})$
    $\widehat{v}_{k+1} = v_k / (1 - \beta_2^{k+1})$
    $\boldsymbol{\theta}_{k+1} = \boldsymbol{\theta}_k - \eta_k \widehat{m}_{k+1} / (\sqrt{\widehat{v}_{k+1}} + \epsilon)$;
**end**
Set $\boldsymbol{\theta} = \boldsymbol{\theta}_K$;
Return $\boldsymbol{\theta}$.

---

## B    PROOF OF THEOREM 4

*Proof.* According to the update rule of vanilla gradient descent, it follows that by mean-value theorem there exists $\widehat{\boldsymbol{\theta}} \in [\boldsymbol{\theta}_k, \boldsymbol{\theta}^*]$ such that

$$
\begin{aligned}
\boldsymbol{\theta}_{k+1} - \boldsymbol{\theta}^* &= \boldsymbol{\theta}_k - \boldsymbol{\theta}^* - \eta_k \nabla f(\boldsymbol{\theta}_k) \\
&= \boldsymbol{\theta}_k - \boldsymbol{\theta}^* - \eta_k \left[ \nabla f(\boldsymbol{\theta}^*) + \nabla^2 f(\widehat{\boldsymbol{\theta}})(\boldsymbol{\theta}_k - \boldsymbol{\theta}^*) \right] \\
&= \left[ \mathbf{I} - \eta_k \nabla^2 f(\widehat{\boldsymbol{\theta}}) \right] (\boldsymbol{\theta}_k - \boldsymbol{\theta}^*),
\end{aligned}
$$

where the last equality holds since $\boldsymbol{\theta}^*$ is a local minimum. By taking the norm, we get

$$
\|\boldsymbol{\theta}_{k+1} - \boldsymbol{\theta}^*\| = \left\| \left[ \mathbf{I} - \eta_k \nabla^2 f(\widehat{\boldsymbol{\theta}}) \right] (\boldsymbol{\theta}_k - \boldsymbol{\theta}^*) \right\| \geq |1 - \eta_k \mu| \, \|\boldsymbol{\theta}_k - \boldsymbol{\theta}^*\|,
$$

where the last inequality holds by our local strong convexity assumption, the fact that $\widehat{\boldsymbol{\theta}} \in \mathbb{B}_\delta \{\boldsymbol{\theta}^*\}$ and our choice of $\eta_k$. The former choice also imply that

$$
\|\boldsymbol{\theta}_{k+1} - \boldsymbol{\theta}^*\| \geq (\eta_k \mu - 1)\|\boldsymbol{\theta}_k - \boldsymbol{\theta}^*\| \geq (1 + \epsilon)\|\boldsymbol{\theta}_k - \boldsymbol{\theta}^*\|,
$$

which yields

$$
\|\boldsymbol{\theta}_k - \boldsymbol{\theta}^*\| \geq (1 + \varepsilon)^k \|\boldsymbol{\theta}_0 - \boldsymbol{\theta}^*\|.
$$

Let $\|\boldsymbol{\theta}_0 - \boldsymbol{\theta}^*\| = D$ and $\widehat{k} = \dfrac{1}{\log(1 + \varepsilon)} \log\left(\dfrac{\delta}{D}\right)$, then

$$
\|\boldsymbol{\theta}_{\widehat{k}} - \boldsymbol{\theta}^*\| \geq \delta,
$$

which completes our proof.

$\square$

## C    PROOF OF THEOREM 5

*Proof.* According to Lemma 4, running vanilla gradient descent with $\eta_k \geq \dfrac{2 + \varepsilon}{\mu}$ for some fixed $\varepsilon > 0$ escapes the neighborhood $\mathbb{B}_\delta(\boldsymbol{\theta}^*)$. Hence, to complete our proof, it suffices to show that GD-SALR will dynamically choose a sufficiently large step size.

We first start by computing a lower bound for our local sharpness approximation in local strongly convex regions. By definition,

$$
\widehat{S}_k = f\left(\boldsymbol{\theta}_{k,+}^{(n_2)}\right) - f\left(\boldsymbol{\theta}_{k,-}^{(n_1)}\right) = f\left(\boldsymbol{\theta}_{k,+}^{(n_2)}\right) - f(\boldsymbol{\theta}_k) + f(\boldsymbol{\theta}_k) - f\left(\boldsymbol{\theta}_{k,-}^{(n_1)}\right).
$$

We start by computing a lower bound for $f(\boldsymbol{\theta}_k) - f\left(\boldsymbol{\theta}_{k,-}^{(n_1)}\right)$. By descent lemma (Bertsekas, 1997),

$$
\begin{aligned}
f\left(\boldsymbol{\theta}_{k,-}^{(i+1)}\right) &\leq f\left(\boldsymbol{\theta}_{k,-}^{(i)}\right) + \left\langle \nabla f\left(\boldsymbol{\theta}_{k,-}^{(i)}\right), \boldsymbol{\theta}_{k,-}^{(i+1)} - \boldsymbol{\theta}_{k,-}^{(i)} \right\rangle + \frac{L}{2}\left\|\boldsymbol{\theta}_{k,-}^{(i+1)} - \boldsymbol{\theta}_{k,-}^{(i)}\right\|^2 \\
&= f\left(\boldsymbol{\theta}_{k,-}^{(i)}\right) - \gamma \left\|\nabla f\left(\boldsymbol{\theta}_{k,-}^{(i)}\right)\right\| + \frac{L\gamma^2}{2}.
\end{aligned}
$$

By summing over the $n_1$ iterations, we get

$$
f(\boldsymbol{\theta}_k) - f\left(\boldsymbol{\theta}_{k,-}^{(n_1)}\right) \geq \gamma \sum_{i=0}^{n_1-1} \left\|\nabla f\left(\boldsymbol{\theta}_{k,-}^{(i)}\right)\right\| - \frac{n_1 L \gamma^2}{2}. \tag{4}
$$

By mean value theorem, there exists $\mathbf{z}_{k,-}^{(i)} \in \left[\boldsymbol{\theta}_{k,-}^{(i)}, \boldsymbol{\theta}_{k,-}^{(i+1)}\right]$ with

$$
\begin{aligned}
\nabla f\left(\boldsymbol{\theta}_{k,-}^{(i+1)}\right) &= \nabla f\left(\boldsymbol{\theta}_{k,-}^{(i)}\right) + \nabla^2 f\left(\mathbf{z}_{k,-}^{(i)}\right)\left(\boldsymbol{\theta}_{k,-}^{(i+1)} - \boldsymbol{\theta}_{k,-}^{(i)}\right) \\
&= \nabla f\left(\boldsymbol{\theta}_{k,-}^{(i)}\right) - \gamma \nabla^2 f\left(\mathbf{z}_{k,-}^{(i)}\right) \frac{\nabla f\left(\boldsymbol{\theta}_{k,-}^{(i)}\right)}{\left\|\nabla f(\boldsymbol{\theta}_{k,-}^{(i)})\right\|},
\end{aligned}
$$

which yields

$$\left\| \nabla f\left(\boldsymbol{\theta}_{k,-}^{(i+1)}\right)\right\| \leq (1 - \gamma\mu/g_{max}) \left\|\nabla f\left(\boldsymbol{\theta}_{k,-}^{(i)}\right)\right\| = \left(\frac{L\, g_{max} - \mu\, g_{min}}{L\, g_{max}}\right) \left\|\nabla f\left(\boldsymbol{\theta}_{k,-}^{(i)}\right)\right\|.$$

Here the inequality holds by local strong convexity and the fact that $\mathbf{z}_{k,-}^i \in \mathbb{B}_\delta(\boldsymbol{\theta}^*)$ due to our choice of $\delta$, the upper bound on the norm of the gradient, and our choice of $\gamma$. Substituting back into equation 4, we get

$$
\begin{aligned}
f(\boldsymbol{\theta}_k) - f\left(\boldsymbol{\theta}_{k,-}^{(n_1)}\right) &\geq \gamma \sum_{i=0}^{n_1-1}\left(\frac{L\, g_{max} - \mu\, g_{min}}{L\, g_{max}}\right)^{-i} \left\|\nabla f\left(\boldsymbol{\theta}_{k,-}^{(n_1-1)}\right)\right\| - \frac{n_1 L \gamma^2}{2} \\
&= \frac{g_{min}}{L}\left(\frac{L\, g_{max}}{\mu g_{min}} - 1\right)\left(\left(\frac{L\, g_{max}}{L\, g_{max} - \mu\, g_{min}}\right)^{n_1} - 1\right)\left\|\nabla f\left(\boldsymbol{\theta}_{k,-}^{(n_1-1)}\right)\right\| - \frac{n_1 L \gamma^2}{2} \\
&= \left(\frac{g_{max}}{\mu} - \frac{g_{min}}{L}\right)\left(\left(\frac{L\, g_{max}}{L\, g_{max} - \mu\, g_{min}}\right)^{n_1} - 1\right)\left\|\nabla f\left(\boldsymbol{\theta}_{k,-}^{(n_1-1)}\right)\right\| - \frac{n_1 L \gamma^2}{2}
\end{aligned}
$$
$$\tag{5}$$

By our choice of $n_1$, we have

$$\left(1 + \frac{\mu g_{min}}{L\, g_{max} - \mu\, g_{min}}\right)^{n_1} - 1 \geq n_1\, a_1. \tag{6}$$

The inequality holds since

$$\log\left(1 + \frac{\mu\, g_{min}}{L\, g_{max} - \mu\, g_{min}}\right)^{n_1} = n_1 \log\left(1 + \frac{\mu\, g_{min}}{L\, g_{max} - \mu\, g_{min}}\right) \geq \frac{n_1\, a_1}{\sqrt{n_1\, a_1 + 1}} \geq \log(1 + n_1 a_1),$$

where the first inequality holds by our choice of $n_1$ and the second inequality is an upper bound of $\log(1 + x)$. By substituting equation 6 in equation 5 and using our assumption that

$$\min\left\{\left\|\nabla f\left(\boldsymbol{\theta}_{k,-}^{(n_1-1)}\right)\right\|, \left\|\nabla f\left(\boldsymbol{\theta}_{k,+}^0\right)\right\|\right\} \geq g_{min},$$

we get

$$
\begin{aligned}
f(\boldsymbol{\theta}_k) - f\left(\boldsymbol{\theta}_{k,-}^{(n_1)}\right) &\geq \left(\frac{g_{max}}{\mu} - \frac{g_{min}}{L}\right) n_1\, a_1\, g_{min} - \frac{n_1\, g_{min}^2}{2L} \\
&\geq n_1 \left(\frac{2 + \epsilon}{\eta_0 \mu}\right)\left(\frac{L_0\, g_{min}}{L}\right).
\end{aligned}
$$
$$\tag{7}$$

We now compute a lower bound for $f\left(\boldsymbol{\theta}_{k,+}^{(n_2)}\right) - f(\boldsymbol{\theta}_k)$. By local strong convexity of $f$, we have

$$
\begin{aligned}
f\left(\boldsymbol{\theta}_{k,+}^{(i+1)}\right) &\geq f\left(\boldsymbol{\theta}_{k,+}^{(i)}\right) + \left\langle \nabla f\left(\boldsymbol{\theta}_{k,+}^{(i)}\right), \boldsymbol{\theta}_{k,+}^{(i+1)} - \boldsymbol{\theta}_{k,+}^{(i)}\right\rangle + \frac{\mu}{2}\left\|\boldsymbol{\theta}_{k,+}^{(i+1)} - \boldsymbol{\theta}_{k,+}^{(i)}\right\|^2 \\
&= f\left(\boldsymbol{\theta}_{k,+}^{(i)}\right) + \gamma\left\|\nabla f\left(\boldsymbol{\theta}_{k,+}^{(i)}\right)\right\| + \frac{\mu\gamma^2}{2}.
\end{aligned}
$$

By summing over the $n_2$ iterations, we get

$$f\left(\boldsymbol{\theta}_{k,+}^{(n_2)}\right) - f(\boldsymbol{\theta}_k) \geq \gamma \sum_{i=0}^{n_2-1}\left\|\nabla f\left(\boldsymbol{\theta}_{k,+}^i\right)\right\| + \frac{n_2 \mu\gamma^2}{2}. \tag{8}$$

By mean value theorem, there exists $\mathbf{z}_{k,+}^i \in \left[\boldsymbol{\theta}_{k,+}^i, \boldsymbol{\theta}_{k,+}^{(i+1)}\right]$ with

$$
\begin{aligned}
\nabla f\left(\boldsymbol{\theta}_{k,+}^{(i+1)}\right) &= \nabla f\left(\boldsymbol{\theta}_{k,+}^{(i)}\right) + \nabla^2 f\left(\mathbf{z}_{k,+}^i\right)\left(\boldsymbol{\theta}_{k,+}^{(i+1)} - \boldsymbol{\theta}_{k,+}^{(i)}\right) \\
&= \nabla f\left(\boldsymbol{\theta}_{k,+}^{(i)}\right) + \gamma \nabla^2 f\left(\mathbf{z}_{k,+}^i\right)\frac{\nabla f\left(\boldsymbol{\theta}_{k,+}^{(i)}\right)}{\left\|\nabla f\left(\boldsymbol{\theta}_{k,+}^{(i)}\right)\right\|},
\end{aligned}
$$

which yields
$$\left\|\nabla f\left(\boldsymbol{\theta}_{k,+}^{(i+1)}\right)\right\| \geq (1 + \gamma\mu/g_{max})\left\|\nabla f\left(\boldsymbol{\theta}_{k,+}^{(i)}\right)\right\|.$$

Here the inequality holds by local strong convexity and the fact that $\mathbf{z}_{k,+}^i \in \mathbb{B}_\delta(\boldsymbol{\theta}^*)$, the upper bound on the norm of the gradient, and our choice of $\gamma$. Substituting back into equation 8, we get

$$
\begin{aligned}
f\left(\boldsymbol{\theta}_{k,+}^{(n_2)}\right) - f(\boldsymbol{\theta}_k) &\geq \gamma \sum_{i=0}^{n_2-1} (1 + \gamma\mu/g_{max})^i \left\|\nabla f\left(\boldsymbol{\theta}_{k,+}^0\right)\right\| + \frac{n_2\mu\gamma^2}{2} \\
&= \frac{g_{max}}{\mu}\left(\left(1 + \frac{\mu g_{min}}{L\,g_{max}}\right)^{n_2} - 1\right)\left\|\nabla f\left(\boldsymbol{\theta}_{k,+}^0\right)\right\| + \frac{n_2\mu\gamma^2}{2}
\end{aligned}
\tag{9}
$$

By our choice of $n_2$, we have

$$\left(1 + \frac{\mu g_{min}}{L\,g_{max}}\right)^{n_2} - 1 \geq n_2\,a_2. \tag{10}$$

The inequality holds since

$$\log\left(1 + \frac{\mu g_{min}}{L\,g_{max}}\right)^{n_2} = n_2\log\left(1 + \frac{\mu g_{min}}{L\,g_{max}}\right) \geq \frac{n_2 a_2}{\sqrt{n_2 a_2 + 1}} \geq \log(1 + n_2 a_2),$$

where the first inequality holds by our choice of $n_2$ and the second inequality is an upper bound of $\log(1 + x)$. By substituting equation 10 in equation 9 and using our assumption that

$$\min\left\{\left\|\nabla f\left(\boldsymbol{\theta}_{k,-}^{(n_1-1)}\right)\right\|, \left\|\nabla f\left(\boldsymbol{\theta}_{k,+}^0\right)\right\|\right\} \geq g_{min},$$

we get

$$
\begin{aligned}
f\left(\boldsymbol{\theta}_{k,+}^{(n_2)}\right) - f(\boldsymbol{\theta}_k) &\geq \frac{g_{max}n_2 a_2\,g_{min}}{\mu} + \frac{n_2\mu g_{min}^2}{2L^2} \\
&\geq n_2\left(\frac{2+\epsilon}{\mu\,\eta_0}\right)\left(\frac{L_0\,g_{min}}{L}\right).
\end{aligned}
\tag{11}
$$

By adding equation 7 and equation 11, we obtain

$$\widehat{S}_k \geq (n_1 + n_2)\left(\frac{2+\epsilon}{\mu\,\eta_0}\right)\left(\frac{L_0\,g_{min}}{L}\right). \tag{12}$$

We now provide an upper bound for $\text{Median}\left(\widehat{S}_k\right)$. Using the Lipschitz property of function, we have

$$f\left(\boldsymbol{\theta}_k\right) - f\left(\boldsymbol{\theta}_{k,-}^{(n_1)}\right) = \sum_{i=0}^{n_1-1} f\left(\boldsymbol{\theta}_{k,-}^{(i)}\right) - f\left(\boldsymbol{\theta}_{k,-}^{(i+1)}\right) \leq L_0 \sum_{i=0}^{n_1-1}\left\|\boldsymbol{\theta}_{k,-}^{(i)} - \boldsymbol{\theta}_{k,-}^{(i+1)}\right\| = n_1 L_0\gamma. \tag{13}$$

$$f\left(\boldsymbol{\theta}_{k,+}^{(n_2)}\right) - f\left(\boldsymbol{\theta}_k\right) = \sum_{i=0}^{n_2-1} f\left(\boldsymbol{\theta}_{k,+}^{(i+1)}\right) - f\left(\boldsymbol{\theta}_{k,+}^{(i)}\right) \leq L_0 \sum_{i=0}^{n_2-1}\gamma\left\|\boldsymbol{\theta}_{k,+}^{(i+1)} - \boldsymbol{\theta}_{k,+}^{(i)}\right\| = n_2\,L_0\gamma. \tag{14}$$

Combining equation 13 and equation 14, we get

$$\mathcal{S} = \text{Median}\left(\widehat{S}_k\right) \leq (n_1 + n_2)L_0\gamma = (n_1 + n_2)\frac{g_{min}L_0}{L}. \tag{15}$$

According to the definition of our learning rate, combining equation 12 and equation 15 results in the following inequality

$$\eta_k = \eta_0\frac{\mathcal{S}_k}{\mathcal{S}} \geq \frac{2+\epsilon}{\mu}. \tag{16}$$

The proof is concluded using Theorem 4. $\qquad\square$

## D    EXPERIMENTAL SETTING

In this section, we provide the detailed experiment settings. All methods are trained with batch normalization and dropout with probability 0.5 after each layer. The batch size is 128. The base learning rate is set to 0.01. Both batch size and learning rate can be adjusted but their ratio should remain constant as suggested in Smith & Le (2017); Smith et al. (2018).

### D.1    MNIST

1. SGD: we train SGD for 100 epochs with a learning rate 0.01 that drops by a factor of 10 after every 30 epochs.

2. SWA: in the first 75 epochs, we run the regular SGD. We then switch to a cyclic learning rate schedule with $\alpha_1 = 5 \times 10^{-3}$ and $\alpha_2 = 1 \times 10^{-4}$, where $\alpha_1$ is the initial learning rate within a cycle and $\alpha_2$ is the ending learning rate within a cycle.

3. Entropy-SGD: we train Entropy-SGD for 20 epochs with $L = 5$. The learning rate for the stochastic gradient Langevin dynamics (SGLD) is set to 0.1 with thermal noise $10^{-4}$. The initial value of the scope is set to 0.03 which increases by a factor of 1.001 after each parameter update.

4. Entropy-SGD-SALR: the learning rate for Entropy-SGD is updated based on Algorithm 2 ($c = 2, \gamma = 0.002$).

5. SGD-SALR: we use base learning rate 0.01 and set $c = 2, \gamma = 0.002$.

### D.2    CIFAR-10

1. SGD: we train SGD for 200 epochs with a learning rate 0.01 that drops by a factor of 10 after every 30 epochs.

2. SWA: in the first 150 epochs, we run the regular SGD. We then switch to a cyclic learning rate schedule with $\alpha_1 = 5 \times 10^{-3}$ and $\alpha_2 = 1 \times 10^{-4}$.

3. Entropy-SGD: we train Entropy-SGD for 40 epochs with $L = 5$. The learning rate for the SGLD is set to 0.1 with thermal noise $10^{-4}$. The initial value of the scope is set to 0.03 which increases by a factor of 1.001 after each parameter update.

4. Entropy-SGD-SALR: the learning rate for Entropy-SGD is updated based on Algorithm 2 ($c = 2, \gamma = 0.002$).

5. SGD-SALR: we use base learning rate 0.01 and set $c = 2, \gamma = 0.002$.

## E    MORE EXPERIMENTS

In this section, we have added more experimental results to illustrate the performance of SALR.

1. We train an LSTM network on the Penn Tree Bank (PTB) dataset for word-level text pre-diction. Following the guideline in Zaremba et al. (2014) and Chaudhari et al. (2019), we train PTB-LSTM with 66 million weights. SGD and SWA are trained with 55 epochs. Entropy-SGD ($L = 5$) and SALR ($c = 2$) are trained with 11 epochs. Overall, **all methods have the same number of gradient calls (i.e., wall-clock times).** We report the word-level perplexity on the test set. Both Entropy-SGD and SALR can obtain lower perplexities with much fewer training epochs.

| PTB | SGD | SWA | Entropy-SGD | SALR |
|---|---|---|---|---|
| PTB-LSTM | 78.4 (0.22) | 78.1 (0.25) | 72.15 (0.16) | 71.42 (0.14) |

Table 6: Perplexity on PTB

2. We train an LSTM to perform character-level text-prediction using War and Peace (WP). We follow the procedures in Chaudhari et al. (2019) and Karpathy et al. (2015). We train Adam/SWA and Entropy-SGD/SALR with 50 and 10 epochs, respectively. Overall, **all methods have the same number of gradient calls.**

| WP | SGD | SWA | Entropy-SGD | SALR |
|---|---|---|---|---|
| LSTM | 1.223 (0.01) | 1.220 (0.05) | 1.095 (0.01) | 1.089 (0.02) |

Table 7: Perplexity on War and Peace

3. We add two experiments on CIFAR-10 and CIFAR-100.

| ResNet56 | SGD | SWA | Entropy-SGD | SALR |
|---|---|---|---|---|
| CIFAR-10 | 93.25 (0.04) | 93.33 (0.02) | 94.17 (0.01) | 94.35 (0.01) |
| CIFAR-100 | 74.75 (0.02) | 74.72 (0.01) | 75.29 (0.01) | 75.30 (0.01) |

Table 8: More Results on CIFAR-10/100

4. We compare SALR with SmoothOut (Wen et al., 2018), a technique to smooth out sharp minima by averaging over multiple perturbed copies of the landscape. We run SALR, SmoothOut and AdamSmoothOut on CIFAR-10 and CIFAR-100 five times using ResNet 44.

| ResNet44 | SmoothOut | AdamSmoothOut | SALR |
|---|---|---|---|
| CIFAR-10 | 92.02 (0.02) | 92.15 (0.03) | 92.45 (0.01) |
| CIFAR-100 | 68.70 (0.02) | 69.21 (0.02) | 70.35 (0.03) |

Table 9: SmoothOut

In conclusion, SALR can deliver improvement over a range of dataset.

