# OpenReview forum: "SALR: Sharpness-aware Learning Rates for Improved Generalization"
_ICLR.cc/2021/Conference — Reject_

### Official Review · AnonReviewer1 · 2020-10-16
**showing some nice empirical results**

**Rating:** 6
**Confidence:** 3

**Review:**

This work proposes an algorithm that aims at finding a flat minimizer. The high-level strategy in the design of the proposed algorithm is increasing the learning rate when the iterate is in the region of a sharp minimizer. The authors claim that by increasing the learning rate, the iterate can get out of the undesired region. To estimate the local landscape (local sharpness), the authors provide a heuristic, which requires running gradient descent and gradient ascent for some number of iterations. The proposed algorithm has a promising result empirically, compared to entropy SGD (Chaudhari et al.) which also aims at finding a solution that generalizes well.

Strength:

(1) Paper is written well. The motivation is clearly explained.

(2) The algorithm is easy to implement.

(3) The algorithm seems to work well in practice.

Weakness:

(1) Compared to the empirical results shown in the paper, the theoretical result is relatively weak. My concern is that Theorem 5 does not differentiate between flat and sharp local minima. Specifically, it could be the case that the conditions of the theorem are satisfied and that the iterate generated by the proposed algorithm escapes flat minima eventually.

The paper will be in a much better shape if the theoretical result shows that the iterate escapes sharp local minima while attracts to flat minima.

(2) (Computational overhead) Since the algorithm has a subroutine to estimate the local landscape, it incurs computational overhead compared to the baselines. I hope the authors can discuss this issue.

Minor:

(1) (Assumption 1) What are the examples that satisfy the assumption?

(2) (Remark 3 and normalization in Definition 1)
I agree with Remark 3, but the argument made in Remark 3 also implies that the normalization will reduce the sharpness value when the gradient is large, which might be an undesired effect.

===
Overall, I think this is a reasonable paper and I am happy to increase the score if the authors respond well.

=== after rebuttal ===

I thank the authors for the feedback.

Regarding the proposed theorem, I was hoping to see if the iterate can stay in the flat minima instead of leaving the region eventually, as the constant of the local strong convexity is inside the log's, which is not very sensitive to the value (the landscape) and might be tricky to provide useful guidance to differentiate flat and sharp minima.

I decided to maintain my score, but I still recommend a weak accept of this paper.

---

> ### Author Response · Authors · 2020-11-16
> **To Reviewer 1**
>
> Dear Reviewer 1, thanks for your feedback. We have tried our best to address all your concerns during the rebuttal period. Our work tries to propose an alternative and first of its kind perspective on improving generalization - one that tackles flatness heads-on via learning rates. Further, a central advantage is that learning rates are generated automatically through the algorithm and refined with iterations.
>
> We have added experiments on CIFAR-100 and two language models to illustrate the performance of SALR. Please check the details below.
>
> (1) We train an LSTM network on the Penn Tree Bank (PTB) dataset for word-level text prediction. Following the guideline in [1] and [2], we train PTB-LSTM with 66 million weights. SGD and SWA are trained with 55 epochs. Entropy-SGD and SALR are trained with 11 epochs. Overall, all methods have the same number of gradient calls (i.e., wall-clock times). We report the word-level perplexity on the test set. As you can see, both Entropy-SGD and SALR can obtain lower perplexities with much fewer training epochs.
>
> Data: PTB, Network: PTB-LSTM, SGD: 78.4 (0.22), SWA: 78.1 (0.25), Entropy-SGD: 72.15 (0.16), SALR: 71.42 (0.14).
>
> (2) We train an LSTM to perform character-level text-prediction using War and Peace (WP). We follow the procedures in [2] and [3]. We train Adam/SWA and Entropy-SGD/SALR with 50 and 10 epochs, respectively. Overall, all methods have the same number of gradient calls.
>
> Data: WP, Network: LSTM, SGD: 1.223 (0.01), SWA: 1.220 (0.05), Entropy-SGD: 1.095 (0.01), SALR: 1.089 (0.02).
>
> (3) Finally, we add two experiments on CIFAR-10 and CIFAR-100.
>
> Data: CIFAR-100, Network: ResNet56, SGD: 74.75 (0.02), SWA: 74.72 (0.01), Entropy-SGD: 75.29 (0.01), SALR: 75.30 (0.01).
>
> Data: CIFAR-10, Network: ResNet56, SGD: 93.25 (0.04), SWA: 93.33 (0.02), Entropy-SGD: 94.17 (0.01), SALR: 94.35 (0.01).
>
>
> We also updated our main paper. Please see Section 6 in the main paper and Section E in the Appendix for details. Overall, SALR yields significant improvement over state of the art techniques aimed at better generalization including SWA and Entropy-SGD, on a broad range of datasets. It also provides an automatic way of defining learning rates that requires no pre-specifications or tuning.
>
> Weakness (1): Our theorem states that when the function is strongly convex around a local minimizers (i.e. local minimizer is sharp), GD-SALR can escape the neighborhood by choosing a large enough step-size. Hence, our methods differentiates between flat and sharp local minima. In fact, higher $\mu$ (strong convexity parameter) reflects sharper minimizers. Our result indeed shows that as $\mu$ increases we require a lower $n_1$ and $n_2$ steps or lower $k$ to escape the neighborhood.
>
> Weakness (2): Yes you are right. However, SALR requires much fewer epoch to achieve a comparable performance with SGD, SWA and others. For example, as shown in Figure 4, SALR only requires 40 epochs while SGD requires 200 epochs. In our experiment, we ensure all models have the exactly same number of gradient calls such that the wall clock times are similar. The Entropy-sgd paper also provides a similar argument. Overall, though sharpness estimation incurs extra computational cost, SALR requires much fewer epochs. We will further emphasis this in the paper.
>
> Minor (1): Assumption 1 is very common in many deep learning papers [1-3]. It mainly assume the function is smooth and gradient is bounded.
>
> Minor (2):  We totally agree, however this helps tune the step size for the gradient ascent and descent steps by standardizing each step.
>
> We hope that your concerns have been successfully addressed and kindly hope you can re-evaluate our paper.
>
> [1] Ge, Rong, et al. "Escaping from saddle points—online stochastic gradient for tensor decomposition." Conference on Learning Theory. 2015.
>
> [2] Jin, Chi, et al. "How to escape saddle points efficiently." arXiv preprint arXiv:1703.00887 (2017).
>
> [3] Kleinberg, Robert, Yuanzhi Li, and Yang Yuan. "An alternative view: When does SGD escape local minima?." ICML (2018).

---

### Official Review · AnonReviewer4 · 2020-10-28
**Official Blind Review #4**

**Rating:** 6
**Confidence:** 3

**Review:**

This paper proposes a method called SALR, which adeptly updates the learning rate based on the local sharpness of the loss function to escape the sharp local minimum. By doing this, the author shows that their method can enhance the generalization and converge speed during the training procedure by encouraging the network to converge to a flat minimum.

The paper provides theoretical analysis as well as empirical results to support their claim.  For the theory part, the paper gives the evidence to prove that SALR can get a better converge rate while in the empirical part, the authors test SALR on MNIST and CIFAR10 using several different network structures.

Here are the pros and Cons I think in these parts.

Pros:
1. The analysis of the converge rate part looks good for me. Though I do not check the proof one-by-one.
2. The experiment covers different networks and different optimizers, which greatly shows the algorithms generalization ability in most of the gradient-based optimizer.

Cons:
1. When calculating stochastic sharpness in empirical using algorithm 2, it seems that we need additional n1 and n2 steps. What is the value of n1 and n2 in empirical? This can slow down the gradient calculation speed a lot, I am worrying this can cause the algorithm very expensive to apply in many cases.

2. The empirical experiment result still seems limited.
    (1). The author only tests several large networks on MNIST and CIFAR10, which makes its generalization statement weak. Moreover, I notice that the author actually does not even test CIFAR10 on MobileNetV2.
    (2). For the statement of faster converge speed, I can not actually see the obvious change from the empirical result from figure 4.

Suggestions:
1. Could you add more experiments on other benchmarks like CIFAR100 and even ImageNet? If it is hard, could you clarify what is the bottleneck?

2. It should be good to see more ablation studies towards the converging speed in the empirical experiment.


Overall, I think this paper gives interesting points with the theoretical analysis. However, I still think the algorithm is hard to convince me in the empirical parts.

============================== Update after author's response ==============================

The additional experiments look good to me, it solves most of my concern, I would like to lift the score to 6.

---

> ### Author Response · Authors · 2020-11-16
> **To Reviewer 4**
>
> Dear Reviewer 4, thanks for your feedback. We have tried our best to address all your concerns during the rebuttal period.
>
> We have added experiments on CIFAR-100 and two language models to illustrate the performance of SALR. Please check the details below. We also updated our main paper. Please see Section 6 in the main paper and Section E in the Appendix for details. Overall, SALR yields significant improvement over state of the art techniques aimed at better generalization including SWA, Entropy-SGD and SmoothOut, on a broad range of datasets. It also provides an automatic way of defining learning rates that requires no pre-specifications or tuning.
>
> Cons (1): As mentioned in section 6, we set $n_1=n_2=5$ and calculate the sharpness measure every $c=2$ iterations. Therefore, SALR requires 5 times more gradient calls at each iteration than SGD. However, SALR requires much fewer epoch to achieve a comparable performance with SGD, SWA and others. As shown in Figure 4, SALR only needs 40 epochs while SGD requires 200 epochs. In our experiment, we ensure all models have the exactly same number of gradient calls such that the running time is similar. The Entropy-sgd paper also provides a similar argument.
>
> Cons (2)/Suggestions: Thanks for pointing it out. We have added more experiments. Specifically, we test SALR and all other benchmark models on CIFAR-100 and two language models. We have attached some results in the rebuttal (Please see the comment to all reviewers).
>
> Finally, we need to clarify that our algorithm has fast convergence. As you can see from Figure 4, the superior performance of SALR can be achieved with 5 times less epochs compared to SGD, ADAM and SWA.
>
> We hope that your concerns have been successfully addressed and kindly hope you can re-evaluate our paper.

---

> ### Author Response · Authors · 2020-11-24
> **Thank you very much!**
>
> Thanks for updating the score!

---

### Official Review · AnonReviewer3 · 2020-10-29
**Paper Requires More Improvement**

**Rating:** 4
**Confidence:** 4

**Review:**


This paper proposes SALR, a new optimization algorithm which adapts the learning rate to avoid sharp local minimas. This is achieved by computing/approximating the sharpness at different iterations and increasing the learning rate when the sharpness is high.

Empirical tests are performed on MNIST, and CIFAR-10, showing that the proposed method works better as compared to naive SGD, and entropy SGD.

Strong Points:
+ The problem being studied is very important and can have high impact for cases such as large batch training in which getting "trapped" in sharp minima is a problem
+ The paper does a good job of intuitively explaining their algorithm which adds noise in sharp regions by increasing learning rate.



Major Comments:

While the proposed algorithm is interesting but the theoretical and empirical results provided need to be strengthened further to support the claims. In particular, please note the following:

1- The empirical results are very limited. Given that the theoretical results are only for simple convex settings, a much more thorough empirical analysis would be needed to evaluate the proposed method.

2- What happens if you have negative curvature?


3- The theoretical proof provided is for simple convex problems. It is not clear how these results would extend to non-convex settings.

4- What is the overhead of the proposed method in terms of wall clock time?

5- How does the proposed approach compare with existing methods such as cyclical learning rate?

I will look for the rebuttal and will adjust my score accordingly.

Suggestions for improvement:

There needs to be a more thorough evaluation of the paper for more challenging learning tasks such as ImageNet classification, and language modeling with transformers. There also needs to be tests to show when the proposed method fails (for example strict saddle points) and how this can be avoided.

---

> ### Author Response · Authors · 2020-11-16
> **To Reviewer 3**
>
> Dear Reviewer 3, thanks for your feedback. We have tried our best to address all your concerns during the rebuttal period. Our work tries to propose an alternative and first of its kind perspective on improving generalization - one that tackles flatness heads-on via learning rates. Further, a central advantage is that learning rates are generated automatically and adaptively through the algorithm and refined with iterations.
>
> 1.  We have now added experiments on CIFAR-100 and two language models to illustrate the performance of SALR. Please check the details below.
>
> (1) We train an LSTM network on the Penn Tree Bank (PTB) dataset for word-level text prediction. Following the guideline in [1] and [2], we train PTB-LSTM with 66 million weights. SGD and SWA are trained with 55 epochs. Entropy-SGD and SALR are trained with 11 epochs. Overall, all methods have the same number of gradient calls (i.e., wall-clock times). We report the word-level perplexity on the test set. As you can see, both Entropy-SGD and SALR can obtain lower perplexities with much fewer training epochs.
>
> Data: PTB, Network: PTB-LSTM, SGD: 78.4 (0.22), SWA: 78.1 (0.25), Entropy-SGD: 72.15 (0.16), SALR: 71.42 (0.14).
>
> (2) We train an LSTM to perform character-level text-prediction using War and Peace (WP). We follow the procedures in [2] and [3]. We train Adam/SWA and Entropy-SGD/SALR with 50 and 10 epochs, respectively. Overall, all methods have the same number of gradient calls.
>
> Data: WP, Network: LSTM, SGD: 1.223 (0.01), SWA: 1.220 (0.05), Entropy-SGD: 1.095 (0.01), SALR: 1.089 (0.02).
>
> (3) Finally, we add two experiments on CIFAR-10 and CIFAR-100.
>
> Data: CIFAR-100, Network: ResNet56, SGD: 74.75 (0.02), SWA: 74.72 (0.01), Entropy-SGD: 75.29 (0.01), SALR: 75.30 (0.01).
>
> Data: CIFAR-10, Network: ResNet56, SGD: 93.25 (0.04), SWA: 93.33 (0.02), Entropy-SGD: 94.17 (0.01), SALR: 94.35 (0.01).
>
> We also updated our main paper. Please see Section 6 in the main paper and Section E in the Appendix for details. Overall, SALR yields significant improvement over state of the art techniques aimed at better generalization including SWA and Entropy-SGD, on a broad range of datasets. It also provides an automatic way of defining learning rates that requires no pre-specifications or tuning.
>
> (About the Theory) We would like to clarify that our result in theorem 5 is a local result that does not assume the global convexity of the loss function. This will be further highlighted in the updated version of the paper. Under few assumptions, GD-SALR can escape the neighborhood by choosing a large enough step-size. This result supports the motivation built behind our algorithm.
>
> 2. We hope you can elaborate more on this question. Is the question addressing the behavior of our algorithm at local maxima or saddle points? Since we run both gradient ascent and descent to estimate sharpness, SALR can handle local maxima. If we are at saddle points, the situation is complicated. When the learning rate is sufficiently small, [1-2] and [4] show that SGD can escape saddle points. Showing similar results for sharp saddle points can be an interesting future direction.
>
> 3. We did not assume the loss function is convex. Please note that our result is a local result in which we assume the function is locally strongly convex around local minimum $\theta$ [3].
>
> 4. In our experiments, we ensure all algorithms have the same number of gradient calls. Therefore, all algorithms have similar wall-clock times. In fact, SALR requires much fewer epoch to achieve a comparable performance with SGD, SWA and others. For example, as shown in Figure 4, SALR only requires 40 epochs while SGD requires 200 epochs. Overall, though sharpness estimation incurs extra computational cost, SALR requires much fewer epochs. We will further emphasis this in the paper.
>
> 5. Please note that the stochastic weighted averaging (SWA) uses a cyclic learning rate schedule and, as shown in section 6, SALR outperforms SWA. Here we note that learning rates in SWA are predetermined and have no guarantees to escape any local regions. For SALR, learning rates are automatically generated based on the loss surface and are refined as more iterations are obtained.
>
> Finally, thanks for your suggestion. We have added more experiments (Please see Section: To All Reviewers). We hope your concerns have been successfully addressed and kindly hope you can re-evaluate our paper.
>
>
> [1] Ge, Rong, et al. "Escaping from saddle points—online stochastic gradient for tensor decomposition." COLT (2015).
>
> [2] Jin, Chi, et al. "How to escape saddle points efficiently." arXiv preprint arXiv:1703.00887 (2017).
>
> [3] Kleinberg, Robert, Yuanzhi Li, and Yang Yuan. "An alternative view: When does SGD escape local minima?." ICML (2018).
>
> [4] Lee, Jason D., et al. "First-order methods almost always avoid saddle points." Conference on Learning Theory (2017).

---

### Official Review · AnonReviewer2 · 2020-10-29

**Rating:** 5
**Confidence:** 5

**Review:**

The paper proposes a simple method (SALR) to encourage the SGD to converge to flatter minima for better generalization. The basic idea is that it increases the learning rate when the sharpness is high, vice versa, such that SGD can escape sharp regions quickly. The sharpness is measured by the difference between the maximum and minimum found by a local SGD with a few fixed steps.

The paper is well written and easy to follow, and shows some interesting observations, such as:
1. SALR can achieve a comparable accuracy using 5 times less epochs in the outer SGD. It will be interesting to see if increasing the epochs can achieve higher accuracy or not;
2. The learning dynamics shown in Figure 4 (Right), indicating SGD escapes some local optima.

However, there are following issues in the paper:

1. discussion on connections to previous works

  1.1. the sharpness in "Definition 1" is highly correlated to gradient magnitude. When the region is smooth, we can loosely say a larger sharpness means a larger gradient magnitude (see Figure 3). Therefore, SALR uses a larger learning rate when the gradient is large. This is contradictory to Adam/AdaGrad, which decreases the learning rate when the accumulated gradients are large. Please explain.

  1.2. the motivation of increasing learning rate in SALR is to increase noise to escape sharp regions. SmoothOut [1] also injects noise to escape sharp regions, by averaging over the same neural network under different small perturbations of parameters. Please clarify the difference.

2. experiments

  2.1. the baselines in Table 1-3 are underperforming than expected. For LeNet, any SGD can achieve above 99% easily. The accuracy (88.44%) of ResNet-50 on Cifar-10 is also unexpectedly low [2].

  2.2. Assuming SALR converges to flatter and flatter regions as it learns, we should expect the sharpness drops as the iteration goes. This is not observed in Figure 4 Right.

Minor:
The bold highlight should go to "5.33 (0.60)" in the 4th row in Table 4.


[1] https://arxiv.org/abs/1805.07898
[2] https://benchmarks.ai/cifar-10

---

> ### Author Response · Authors · 2020-11-16
> **To Reviewer 2**
>
> Dear Reviewer 2, thanks for your feedback. We have tried our best to address all your concerns during the rebuttal period.
>
> We have added experiments on CIFAR-100 and two language models to illustrate the performance of SALR. Please check the details below. We also updated our main paper. Please see Section 6 and Section E for details. Overall, SALR yields significant improvement over state of the art techniques aimed at better generalization including SWA, Entropy-SGD and SmoothOut, on a broad range of datasets. It also provides an automatic way of defining learning rates that requires no pre-specifications or tuning.
>
> 1. Discussion on connections to previous works.
>
> (1) The purpose of SALR is very different from Adam/AdaGrad. Adam or its variants aim at fast convergence. SALR aims at escaping sharp minima when the sharpness is high and to stay at flat regions otherwise. In other words, SALR aims for a better generalization. This explains why SALR is opposite to Adam. Due to the accumulated gradients and decaying learning rates, Adam can get stuck in sharp minima while SALR does not have this issue. Indeed, there are some papers point out that Adam does not generalize well [3-5]. Interestingly, as shown in Table 3, SALR can improve the performance of ADAM using our learning rate schedule strategy. The reason is that SALR helps ADAM escape from sharp minimizers. Also, due to the opposite effect, ADAM-SALR slightly performs worse than SGD-SALR.
>
> In a nutshell, our work tries to propose an alternative and first of its kind perspective on improving generalization - one that tackles flatness heads-on via learning rates. Further, a central advantage is that learning rates are generated automatically through the algorithm and refined with iterations.
>
> (2) We thank the reviewer for referring to this related work. As you pointed out, SmoothOut smooths out sharp minima by averaging over multiple perturbed copies of the landscape. In contrast, our method does not perturb or modify the landscape and does not incur overhead cost when considering the same number of gradient calls. Our method is a general adaptive learning rate framework which can even be implemented over SmoothOut (by changing the learning rate in equation (11) in [1]). To demonstrate the favorable performance of SALR, we run SALR, SmoothOut and AdamSmoothOut on CIFAR-10 and CIFAR-100 five times using ResNet 44 (Experiments in section IV-B in [1]). Please check details below. These discussions will be included in the manuscript.
>
> Network: ResNet 44,
> Data: CIFAR-10,
> SmoothOut: 92.02 (0.02),
> AdamSmoothOut: 92.15 (0.03),
> SALR: 92.45 (0.01).
>
> Network: ResNet 44,
> Data: CIFAR-100,
> SmoothOut: 68.70 (0.02),
> AdamSmoothOut: 69.21 (0.02),
> SALR: 70.35 (0.03).
>
> 2. Experiments
>
> (1) We do not tune any models in our experiment. This ensures fairness of comparison. All models use the same dropout probability, the same initial learning rate, batch normalization and random initialization. To further support our argument, we use the ResNet56 as a baseline and follow the procedure in [2] to obtain a 93.25\% accuracy on CIFAR-10 and a 74.75\% accuracy on CIFAR-100. We attach this preliminary results in the table below. As you can see, SALR can still deliver improvement. The key is that SALR is a general framework and can be viewed as an add-on to many first-order optimization methods.
>
> Data: CIFAR-100, Network: ResNet56, SGD: 74.75 (0.02), SWA: 74.72 (0.01), Entropy-SGD: 75.29 (0.01), SALR: 75.30 (0.01).
>
> Data: CIFAR-10, Network: ResNet56, SGD: 93.25 (0.04), SWA: 93.33 (0.02), Entropy-SGD: 94.17 (0.01), SALR: 94.35 (0.01).
>
> (2) The overall goal of SALR is to jump out of sharp regions through increasing learning rates. Hence, our algorithm can potentially jump through multiple sharp regions before reaching a flat region in which the algorithm converges. This is further supported by Figure 4 (right). In flat regions, the learning rate will drop to support the convergence of the algorithm.
>
> Finally, thanks for your suggestion. We hope your concerns have been successfully addressed and kindly hope you can re-evaluate our paper.
>
>
>
> [1] Wen, Wei, et al. "SmoothOut: Smoothing out sharp minima to improve generalization in deep learning." arXiv preprint arXiv:1805.07898 (2018).
>
> [2] Lee, HyunJae, Hyo-Eun Kim, and Hyeonseob Nam. "Srm: A style-based recalibration module for convolutional neural networks." Proceedings of the IEEE International Conference on Computer Vision. 2019.
>
> [3] Zhou, Pan, et al. "Towards Theoretically Understanding Why SGD Generalizes Better Than ADAM in Deep Learning." Advances in Neural Information Processing Systems 33 (2020).
>
> [4] Keskar, Nitish Shirish, and Richard Socher. "Improving generalization performance by switching from adam to sgd." arXiv preprint arXiv:1712.07628 (2017).
>
> [5] Wilson, Ashia C., et al. "The marginal value of adaptive gradient methods in machine learning." Advances in neural information processing systems. 2017.

---

### Author Response · Authors · 2020-11-16
**To All Reviewers**

We thank all reviewers for their careful reading and useful comments. We have tried our best to address all your concerns during the rebuttal period.

We have added experiment on CIFAR-100 and two language models to illustrate the performance of SALR. Please check the details below. We also updated our main paper. Please see Section 6 in the main paper and Section E in the Appendix for details. Overall, SALR yields significant improvement over state of the art techniques aimed at better generalization including SWA, Entropy-SGD and SmoothOut, on a broad range of datasets. It also provides an automatic way of defining learning rates that requires no pre-specifications or tuning.

1. We train an LSTM network on the Penn Tree Bank (PTB) dataset for word-level text prediction. Following the guideline in [1] and [2], we train PTB-LSTM with 66 million weights. SGD and SWA are trained with 55 epochs. Entropy-SGD ($L=5$) and SALR ($c=2$) are trained with 11 epochs. Overall, all methods have the same number of gradient calls (i.e., wall-clock times). We report the word-level perplexity on the test set. As you can see, both Entropy-SGD and SALR can obtain lower perplexities with much fewer training epochs.

Data: PTB,
Network: PTB-LSTM,
SGD: 78.4 (0.22),
SWA: 78.1 (0.25),
Entropy-SGD: 72.15 (0.16),
SALR: 71.42 (0.14).

2. We train an LSTM to perform character-level text-prediction using War and Peace (WP). We follow the procedures in [2] and [3]. We train Adam/SWA and Entropy-SGD/SALR with 50 and 10 epochs, respectively. Overall, all methods have the same number of gradient calls.

Data: WP,
Network: LSTM,
SGD: 1.223 (0.01),
SWA: 1.220 (0.05),
Entropy-SGD: 1.095 (0.01),
SALR: 1.089 (0.02).

3. Finally we add two experiments on CIFAR-10 and CIFAR-100.

Data: CIFAR-100,
Network: ResNet56,
SGD: 74.75 (0.02),
SWA: 74.72 (0.01),
Entropy-SGD: 75.29 (0.01),
SALR: 75.30 (0.01).


Data: CIFAR-10,
Network: ResNet56,
SGD: 93.25 (0.04),
SWA: 93.33 (0.02),
Entropy-SGD: 94.17 (0.01),
SALR: 94.35 (0.01).

In conclusion, SALR can deliver improvement over a range of dataset.


[1] Zaremba, Wojciech, Ilya Sutskever, and Oriol Vinyals. "Recurrent neural network regularization." ICLR (2015).

[2] Chaudhari, Pratik, et al. "Entropy-sgd: Biasing gradient descent into wide valleys." Journal of Statistical Mechanics: Theory and Experiment 2019.12 (2019): 124018.

[3] Karpathy, Andrej, Justin Johnson, and Li Fei-Fei. "Visualizing and understanding recurrent networks." ICLR (2015).

---

### Decision · Program_Chairs · 2021-01-07
**Final Decision**

**Decision:**

Reject

**Comment:**

This paper proposes a method to update the learning rate dynamically by increasing it in areas with higher sharpness and decreasing it otherwise. This would the hopefully leads to escaping sharp valleys and better generalization. Authors further provide some related theoretical results and several experiments to show effectiveness of their models.

All reviewers find the proposed method well-motivated, novel and interesting. The paper is well-written and easy to follow. However, both theoretical results and empirical evaluations could be improved significantly:

1- The theoretical results as is provides little to no insight about the algorithm and unfortunately, authors do not discuss the insights from the theoretical results adequately in the paper. See for eg. R1's comments about this.

2- Given that the theoretical results are not strong, the thoroughness in empirical evaluation is important and unfortunately the current empirical results is not convincing. In particular, there are two main areas to improve:

a) Based on the Appendix D, the choice of hyper-parameters seem to be made in an arbitrary way and all models are forced to use the same hyper-parameters. This way, the choice of hyper-parameters could potentially favor one method over the other. A more principled approach is to tune hyper-parameters separately for each method.

b) It looks like the choice of #epochs has been made in an arbitrary way. For all experiments, it would be much more informative to have a figure similar to the left panel of Fig. 4 but with much more #epochs so that reader can clearly see if the benefit of SALR would disappear with longer training or not.

c) Based on the current results, SALR's performance  is on par with that of Entropy-SGD on CIFAR-100 and WP and there is a very small gap between them on CIFAR-10 and PTB. I highly recommend adding ImageNet results to make the empirical section stronger. The other option is to compare against other methods in fine-tuning tasks. That is, take a checkpoint of a trained model on ImageNet and compare SALR with other methods on several fine-tuning tasks.

Given the above issues, my final recommendation is to reject the paper. I want to thank authors for engaging with reviewers during the discussion period and adding several empirical results to the revision. I hope authors would address the above issues as well and resubmit their work.